# Diagnosing transit times on the northwestern North Atlantic continental shelf

Krysten Rutherford[1] and Katja Fennel[1]

[1]Department of Oceanography, Dalhousie University, 1355 Oxford Street, Halifax B3H 4R2, Nova Scotia, Canada

**Correspondence:** Krysten Rutherford (Krysten.Rutherford@dal.ca)

**Abstract.** The circulation in the northwestern North Atlantic Ocean is highly complex, characterized by the confluence of two major western boundary current systems and several shelf currents. Here we present the first comprehensive analysis of transport paths and timescales for the northwestern North Atlantic shelf, which is useful for estimating ventilation rates, describing circulation and mixing, characterizing the composition of water masses with respect to different source regions, and elucidating rates and patterns of biogeochemical processing, species dispersal and genetic connectivity. Our analysis uses dye and age tracers within a high-resolution circulation model of the region, divided into 9 sub-regions, to diagnose retention times, transport pathways, and transit times. Retention times are shortest on the Scotian Shelf ($\sim$3 months), where the inshore and shelf-break branches of the coastal current system result in high along-shelf transport to the southwest, and on the Grand Banks ($\sim$3 months). Larger retention times are simulated in the Gulf of St. Lawrence ($\sim$12 months) and the Gulf of Maine ($\sim$6 months). Source water analysis shows that Scotian Shelf water is primarily comprised of waters from the Grand Banks and Gulf of St. Lawrence, with varying composition across the shelf. Contributions from the Gulf of St. Lawrence are larger at near-shore locations, whereas locations near the shelf break have larger contributions from the Grand Banks and slope waters. Waters from the deep slope have little connectivity with the shelf, because the shelf-break current inhibits transport across the shelf break. Grand Banks and Gulf of St. Lawrence waters are therefore dominant controls on biogeochemical properties, and on setting and sustaining planktonic communities on the Scotian Shelf.

## 1 Introduction

Biogeochemical cycling and property distributions in aquatic systems, including in lakes, estuaries and ocean basins, depend on water transport paths and timescales. Analysis of transport paths and timescales has been used to estimate ventilation rates (Jenkins, 1987; England, 1995; Hohmann et al., 1998; Cao et al., 2009), to describe circulation and mixing (Fine, 1995; Haine et al., 1998; Schlosser et al., 2001; Wunsch, 2002), and to investigate river plume dynamics (Zhang et al., 2010, 2012). These analyses can either be the focus of or complement investigations into biogeochemical processes. For example, Holzer and Primeau (2006) used residence time, a diagnostic of transport timescales, to investigate the role of the global overturning circulation in controlling inter-basin transport and nutrient distributions. Cao et al. (2009) used ventilation timescales to assess the effect of ocean transport on $CO_2$ uptake and how different ventilation rates can affect projected anthropogenic carbon in global ocean models. Laruelle et al. (2013) estimated freshwater residence times in the coastal ocean to better un-

derstand the connection between rivers and coastal waters, and their effect on regional and global air-sea $CO_2$ flux. In a recent study, Xue et al. (2011) used flushing times to discuss carbon export from the continental shelf of the East China Sea. Ho et al. (2017) used residence time estimates to infer carbon export fluxes from a mangrove-dominated estuary. These examples show that transport timescales are widely used measures for the analysis of biogeochemical distributions and fluxes.

Transport paths and timescales can be obtained with the help of tracers, either through direct measurement of appropriate chemical tracers that are injected into aquatic systems deliberately (e.g. Banyte et al., 2013; Ho et al., 2017) or unintentionally (Beining and Roether, 1996; Karstensen and Tomczak, 1998), or with the help of dye and age tracers in numerical models (e.g. England, 1995; Delhez et al., 1999; Banas and Hickey, 2005; Zhang et al., 2010). Due to practical considerations such as cost, the application of chemical tracers for direct measurement remains limited, and numerical tracers are often used. Models can

scale up from sparse measurements through mechanistic representations of the relevant processes and achieve high spatial and temporal resolution. Various methods for determining timescales and quantifying ocean age and ventilation rates in numerical models have been used (Jenkins, 1987; Sarmiento et al., 1990; Thiele and Sarmiento, 1990; England, 1995).

  Through the implementation of dye and age tracers in a high-resolution regional circulation model, this present study aims to determine circulation pathways, transport timescales, and residence times for the northwestern (NW) North Atlantic shelf,

which is characterized by complex circulation and is undergoing rapid changes. Although previous studies have quantified shelf basin particle retention (Rogers, 2015), shelf residence times as part of global studies (Bourgeois et al., 2016; Sharples et al., 2017), and transport times from the St. Lawrence River to the Scotian Shelf (Sutcliffe et al., 1976; Smith, 1989; Shan et al., 2016), this is the first comprehensive analysis of residence times, and transport pathways and timescales in the NW North Atlantic.

The region has experienced rapid oxygen loss over past decades both on the Scotian Shelf (Gilbert et al., 2010) and in bottom waters in the Lower St. Lawrence Estuary in the Gulf of St. Lawrence (Gilbert et al., 2005), with negative effects on the coastal ecosystem (Bianucci et al., 2016; Brennan et al., 2016a). Furthermore, the region's status as a source or sink of atmospheric $CO_2$ is unresolved, with conflicting reports of the Scotian Shelf as a region of net outgassing (Shadwick et al., 2010) and net uptake (Signorini et al., 2013). Coastal upwelling, defined as upwelling of water from below the seasonal theromocline

within 10 km of the coast, has been reported along the southern shore of Nova Scotia during the late summer months due to upwelling-favourable winds along the coast (Petrie et al., 1987; Shan et al., 2016). Shadwick et al. (2010) and Burt et al. (2013) have proposed that upwelling also occurs at the shelf break bringing subsurface slope waters onto the Scotian Shelf to explain the reported outgassing of $CO_2$, but this is counter to the hypothesized behavior of mid- and high-latitude shelves as net sinks (Tsunogai et al., 1999; Cai et al., 2006). Ocean circulation and mixing are important controls on oxygen ventilation and

net air-sea $CO_2$ flux. By elucidating transport pathways and quantifying the associated timescales for different sub-regions in the NW North Atlantic, we aim to improve mechanistic understanding of the physical influences on regional biogeochemical cycling, in particular deoxygenation and net air-sea $CO_2$ flux.

  We follow established methodology for quantifying the related concepts of water age and residence time. Age, $a$, is a local measure, in other words it is unique to each water parcel, and thus recognizes spatial heterogeneity (Delhez et al., 1999;

Monsen et al., 2002). The age of a water parcel is defined as the time since that parcel left a specified point location, sur-

face or volume where its age is set to zero (Delhez et al., 1999) and is a Lagrangian concept since it is a characteristic of the fluid parcel (Deleersnijder et al., 2001). There are two main approaches to calculating age in numerical models: Green's function-based transit time distribution (TTD) theory (Hall and Plumb, 1994; Holzer and Hall, 2000; Haine and Hall, 2002; Hall and Haine, 2002) and constituent-oriented age and residence time theory (CART; Delhez et al., 1999; Deleersnijder et al.,

2001; Delhez et al., 2004; Delhez, 2006; Delhez and Deleersnijder, 2006). TTD is best used for steady flow applications and computes the full spectrum or distribution of transit times in a water parcel using Green's functions (Haine and Hall, 2002), while CART is better suited to time-varying flow, and is especially useful for highly resolved coastal applications. The present study uses CART, which calculates the mean age as a mass-weighted average of the distribution of ages, $c(t, x, \tau)$, present in each water parcel (Delhez et al., 1999). A fuller explanation of CART theory is given in Section 3.1.

Residence time, a complement to age, is also a local measure unique to each water parcel. Local residence time estimates can be averaged to quantify mean residence time (also referred to as flushing time or retention time; Agmon, 1984; Monsen et al., 2002), which applies to the larger control volume. Mean residence time, defined as the time it takes for a water parcel to leave a control volume or source volume, is frequently used to quantify the renewal of water in a defined body of water and can be used to assess the influence of hydrodynamic processes on aquatic systems. Since residence time is a function of both space

and time (by definition, every water parcel has its own unique path and history, each with a different residence time), it is necessary to describe residence time in a finite volume as a distribution (Delhez et al., 2004). From this distribution, the mean residence time, $\tau_R$, can be calculated as the first moment of the residence time distribution (Agmon, 1984; Berezhkovskii et al., 1998). Although knowledge of $\tau_R$ does not identify the underlying physical processes or a system's unique spatial distribution of water retention, it is useful for comparing different regions (Monsen et al., 2002).

In this study, we use a high-resolution circulation model for the NW North Atlantic (Brennan et al., 2016b) that is implemented using the Regional Ocean Modeling System (ROMS; Haidvogel et al., 2008). Within the model domain, we distinguish 9 different sub-regions and track their source-water movements with passive dye tracers and associated age tracers following CART. The simulated dye tracer distributions illustrate how water circulates throughout the domain, and enable quantification of mean residence time in four shelf regions: the Grand Banks, the Gulf of St. Lawrence, the Scotian Shelf, and the Gulf of

Maine. Mass fractions of each of the dyes in the Gulf of St. Lawrence, Scotian Shelf, and Gulf of Maine illustrate how the different source regions are contributing at different locations and on average. Lastly, the age of these dyes indicates how long it takes water to travel throughout the domain. Overall, we found that the shelf water is isolated from the slope water, and that there is little cross-shelf exchange.

## 2  Study Region

The NW North Atlantic (Figure 1), off the eastern coast of Canada, is uniquely located at the junction of the subpolar and subtropical gyres (Loder et al., 1997; Hannah et al., 2001). The circulation is dominated by the southward transport of Arctic waters via the Labrador Current system (Loder et al., 1998; Fratantoni and Pickart, 2007). The outflow from the Denmark and Davis Straits in the north, which make up the Labrador Current, accumulates along the northwestern North Atlantic

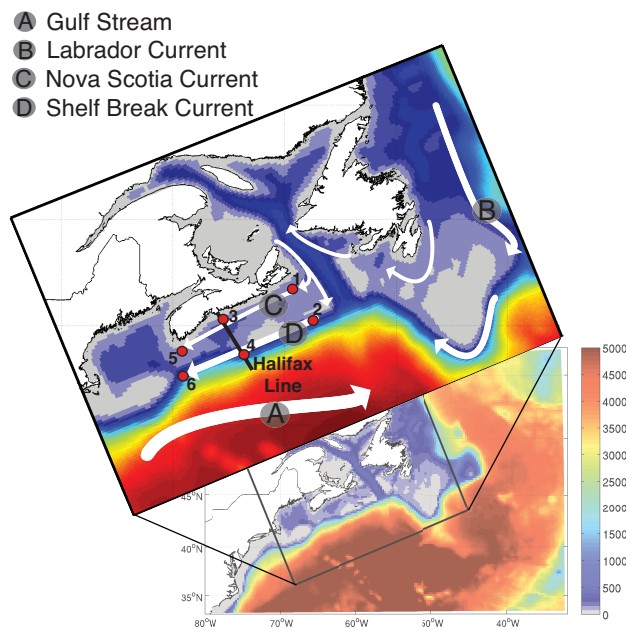

**Figure 1.** Bathymetric map of the NW Atlantic, with the model domain and general circulation structure shown in the upper panel. The Halifax Line, a transect across the Scotian Shelf, is shown by the bold black line. Station locations are numbered and indicated by red points along the Scotian Shelf.

continental shelf separating cold, fresh shelf waters from warm, salty slope waters (Beardsley and Boicourt, 1981; Loder et al., 1998; Fratantoni and Pickart, 2007).

The Scotian Shelf (Figure 2), a 700 km-long portion of the continental shelf centrally located in the NW North Atlantic, is characterized by currents moving to the southwest with inshore and shelf-break branches. The inshore Nova Scotia Current (NSC) originates in the Gulf of St. Lawrence, turns onto the Scotian Shelf at Cabot Strait, moves southwestward along the coast, and enters the Gulf of Maine at Cape Sable. The shelf-break current is an extension of the Labrador Current (Hannah et al., 2001; Han, 2003). Since the NSC plays a dominant role in the alongshore transport on the Scotian Shelf, much of the water within 60 km of shore and top 100 m is composed of Gulf of St. Lawrence-originating water (Dever et al., 2016). Freshwater discharge from the St. Lawrence River additionally impacts the seasonal cycle of the NSC transport (Dever et al., 2016), and both the NSC and shelf break current are strongest in the winter and weakest in the summer (Shan et al., 2016). Overall, this demonstrates the important connectivity between the Gulf of St. Lawrence and the Scotian Shelf. Transit times from the St. Lawrence River (at Quebec City) to Halifax and Cape Sable have been previously estimated as 6 to 7 months (Sutcliffe et al., 1976; Shan et al., 2016) and 8 to 9 months (Smith, 1989; Shan et al., 2016), respectively.

Small-scale circulation patterns on the shelf are influenced by topographic features, such as banks and basins, creating anticyclonic and cyclonic features (Han, 2003). In deep shelf basins and along the shelf break, the contribution from slope

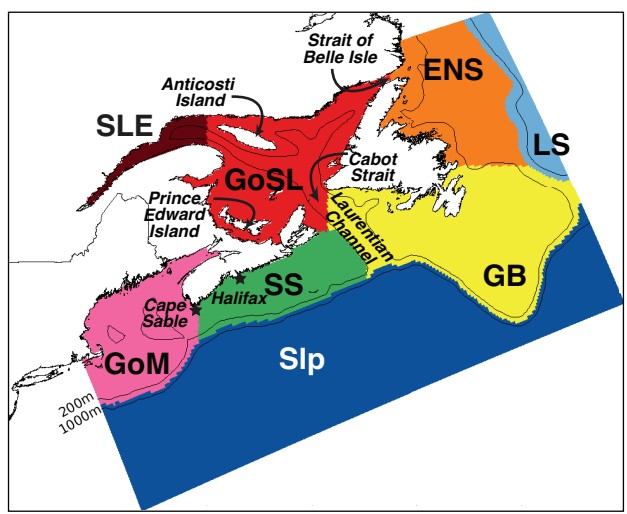

**Figure 2.** Division of the model domain for initialization of dye tracers. Regions are as follows: offshore segment (Slp), which is further divided into 2 depth levels (200 m and above, and below 200 m); Labrador Sea (LS), East Newfoundland Shelf (ENS), Grand Banks (GB), St. Lawrence Estuary (SLE), Gulf of St. Lawrence (GoSL), Scotian Shelf (SS), and Gulf of Maine (GoM).

waters becomes more important (Dever et al., 2016), with onshore flow of slope water through several cross-shelf channels (Shan et al., 2016).

## 3 Methods

### 3.1 CART Theory: Quantifying Age

5  Most methods for quantifying age are based on the idea that in each water parcel there is a distribution of ages, $c(t, x, \tau)$, as shown schematically in Figure 3. All equations to calculate age are based on this age distribution in a water parcel at position $x$ and at time $t$, where $\tau$ is the age or time since the water particle left its source region. The age distribution is subject to the following dynamical equation:

$$\frac{\partial c(\tau)}{\partial t} = -\nabla(\mathbf{u}c - \mathbf{K}\nabla c) - \frac{\partial c(\tau)}{\partial \tau} \tag{1}$$

10  where $\mathbf{u}$ is the velocity at time $t$ and location $x$, and $\mathbf{K}\nabla c$ is the subgrid-scale flux parameterization, with the eddy diffusivity tensor $\mathbf{K}$ (Delhez et al., 1999). The first two terms on the right side represent water being advected and diffused, respectively, which will add and subtract from the age distribution. The last term represents the aging of the distribution (Figure 3).

For coastal applications, such as in the present study, calculating the entire spectrum of ages in Figure 3 is computationally not feasible and instead we calculate the mass-weighted average of the spectrum of ages, i.e. the mean age (CART method). To

15  implement this method in numerical models, two tracers need to be simulated: (1) a passive dye tracer, $C(x, t)$, that is purely

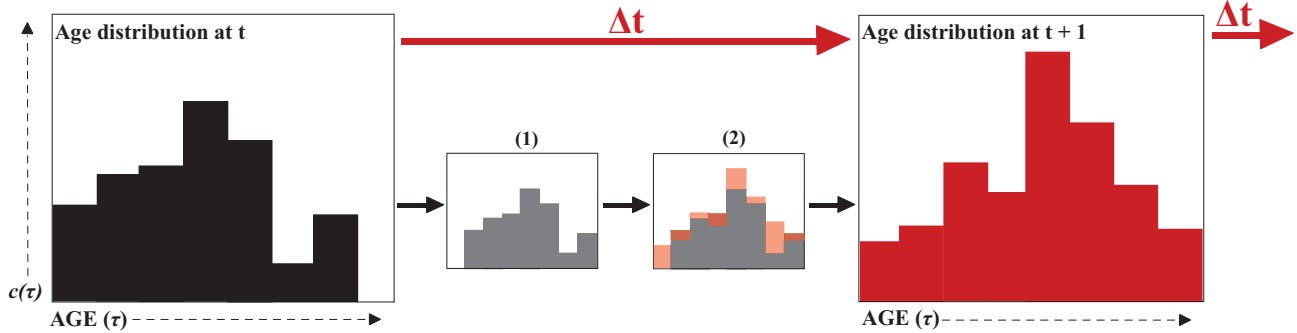

**Figure 3.** Each water parcel is characterized by a distribution of ages, as described in equation (1) and indicated in the left panel. With each time step, this distribution ages as indicated by the shift to small panel (1); additionally, water is advecting/diffusing in/out of the water parcel resulting in different age components being added and subtracted as indicated by the transition from (1) to (2). Together, these two processes produce the distribution at the next time step shown in the right panel.

advected and diffused throughout the domain, and (2) an age concentration tracer, $\alpha(x,t)$, that is coupled with the passive dye tracer and only starts to increase once the coupled tracer has left its source region. This second tracer essentially tracks the time since the associated dye tracer has last left its initialization region and is only defined where the dye tracer is present (i.e. where $C > 0$). Mean age is then calculated as a ratio of these two tracers.

5 Equations governing the dye and age concentration tracers are derived as follows. It can be assumed that all tracer leaves the domain as $\tau$ approaches infinity; in other words, as $\tau$ becomes infinitely large, the age spectrum distribution approaches zero (i.e. $\lim_{\tau \to \infty} c_i(t,x,\tau) = 0$). With this boundary condition, Equation 1 can be integrated with respect to $\tau$ and using

$$C(t,x) = \int_0^\infty c(t,x,\tau)d\tau. \tag{2}$$

The resulting equation

$$10 \quad \frac{\partial C}{\partial t} = -\nabla(\mathbf{u}C - \mathbf{K}\nabla C) \tag{3}$$

is used to track the dye tracer concentration, $C$ (Delhez et al., 1999).

The age concentration tracer, $\alpha(t,x)$, is defined as the first moment of the age distribution, $c(t,x,\tau)$, and is obtained by multiplying Equation 1 by $\tau$, integrating with respect to $\tau$ and making use of the fact that

$$\alpha(t,x) = \int_0^\infty \tau c(t,x,\tau)d\tau \tag{4}$$

15 which yields

$$\frac{\partial \alpha}{\partial t} = C - \nabla(\mathbf{u}\alpha - \mathbf{K}\nabla\alpha). \tag{5}$$

Equation 5 is coupled with Equation 3 such that $\alpha$ will grow proportional in time with each time step when $C > 0$.

The mean age, $a(t,x)$, is defined by the first moment of the age distribution normalized by the dye tracer concentration

$$a(t,x) = \frac{1}{C(t,x)} \int\limits_0^\infty \tau c(t,x,\tau)d\tau. \tag{6}$$

In other words, the age is calculated by dividing the age concentration tracer, $\alpha(t,x)$, by the dye tracer, $C(t,x)$, as in equation
5   6

$$a(t,x) = \frac{\alpha(t,x)}{C(t,x)}. \tag{7}$$

These equations can be applied to any number of independent tracers with separate $C_i(t,x)$, $\alpha_i(t,x)$, and $a_i(t,x)$, as in this study. It is straightforward to implement Equations 3 and 5 in ROMS.

## 3.2   Quantifying Mean Residence Time

Residence time, similar to age, is a local measure described by a distribution. Here, we calculate the mean residence time (MRT), which has also been referred to as "mean passage time" (Agmon, 1984), "average lifetime" (Berezhkovskii et al., 1998), and "flushing time" (Monsen et al., 2002), and describes the average time water or a constituent (e.g. a dissolved tracer) spends in a defined control volume.

Residence time is best defined in a probabilistic framework (Agmon, 1984; Berezhkovskii et al., 1998). The probability that
a particle, $P$, which at time $t_o = 0$ was located at $x_o$ in domain $X$, is found at point $x \in X$ at time $t$, is given by the probability $p(x,t|X)$. Then survival probability, or the probability of $P$ remaining in the domain $X$, is

$$S(t|X) \equiv \int\limits_X p(x,t|X)dx. \tag{8}$$

The lifetime distribution, $F(t|X)$, which is similar to the age distribution and essentially describes how long the particle $P$ spends within the domain $X$, is then described as

$$F(t|X) = -\frac{\partial S(t|X)}{\partial t}. \tag{9}$$

Similar to the calculation of the mean age, the mean lifetime or mean residence time of $P, \tau_P$, is calculated as the first moment of the lifetime distribution

$$\tau_P = -\int\limits_0^\infty \frac{\partial S(t|X)}{\partial t} t dt = \int\limits_0^\infty S(t|X)dt. \tag{10}$$

The same principles used to find the residence time of a single particle can be applied to calculate the MRT of all particles
contained in domain $X$ at $t_o$, e.g. a passive dye tracer homogeneously initialized within $X$. In practice, the lifetime distribution for this scenario is described as the distribution of dye mass, $C_m$, leaving the domain

$$F(t|X) = -\frac{\partial C_m}{\partial t}. \tag{11}$$

MRT is calculated as the first moment of $F(t|X)$ (i.e. the weighted sum of the dye tracer mass leaving the control domain at each time step), integrating to the end of the simulation, $t_n$,

$$\tau_R = -\int_0^{t_n} \frac{\partial C_m}{\partial t} t dt. \tag{12}$$

MRT is different from residence time and age, which are local quantities. Its calculation requires only the implementation of a
passive dye tracer.

### 3.3   Model Domain and Setup

Our model is based on ROMS version 3.5, a terrain-following, free-surface, primitive equation ocean model (Haidvogel et al., 2008), implemented with 30 vertical levels and approximately 10 km horizontal resolution ($240 \times 120$ horizontal grid cells). The model domain includes the Gulf of Maine, Scotian Shelf, East Newfoundland Shelf, Grand Banks and Gulf of St. Lawrence
(see Figure 2), and is described in detail in Brennan et al. (2016b), who have shown it to realistically represent the regional circulation patterns. Our implementation uses the GLS vertical mixing scheme (Umlauf and Burchard, 2003; Warner et al., 2005), atmospheric surface forcing from the European Centre for Medium-Range Weather Forecasts (ECMWF) ERA-Interim global atmospheric reanalysis (Dee et al., 2011), and the *h*igh-order *s*patial *i*nterpolation at the *m*iddle *t*emporal level (HSIMT) advection scheme for tracers (Wu and Zhu, 2010). HSIMT is flux-based, 3rd-order accurate, mass-conservative, oscillation-free,
and positive-definite with low dissipation and no overshooting. Boundary conditions for temperature, salinity and transport are defined using Urrego-Blanco and Sheng's (2012) regional physical ocean model of the NW North Atlantic (their model domain is shown as background in Figure 1), with open boundary transports augmented by barotropic tides from Egbert and Erofeeva (2002). The model's temperature and salinity is nudged toward Urrego-Blanco and Sheng's (2012) solution in a 10-grid-cell-wide buffer zone along the open boundaries. Nudging strength decays linearly away from the boundary to a value of zero at the
11th grid cell. Our model is able to resolve mesoscale features in the region and captures the reported coastal upwelling (see Supplementary information for further details; Petrie et al., 1987; Shan et al., 2016). Our model simulation with dye tracers is initialized on January 1, 1999 from Urrego-Blanco and Sheng's (2012) solution and run for 6 to 9 years.

### 3.4   Numerical Dye and Age Tracer Setup

Numerical dyes are used to mark the 8 chosen sub-regions (Figure 2). The offshore segment (Slp) is divided into two layers
(200 m and above defined as Slp-S, and below 200 m defined as Slp-D), each with its associated dye. In each sub-region, the initial dye concentration is set to 1 kg m$^{-3}$ and the initial age tracer is 0 kg s m$^{-3}$. Two types of numerical tracer experiments are performed. The first type, referred to as TRANS, qualitatively shows the circulation patterns and transit pathways, and is used for the quantification of residence times in selected sub-regions of the model domain. Here dyes are initialized only once at the beginning of the simulation, allowed to advect and diffuse throughout the domain, and their concentration decays
within the source region over the 6-year simulation. The second type of experiment, referred to as AGE, is used to calculate dye mass fractions and ages, and is run for 9 years to ensure that a dynamic steady state is reached for the dye and age tracers.

Here dyes are constantly forced to the initial value in their source regions, effectively re-initializing the dyes at every time step and forcing their ages to zero at the source. Age only starts to increase when the dye leaves the source region, allowing the analysis of age at any point within the domain and the calculation of mean age in sub-regions that are fully contained within the model domain, i.e. the GoSL, SS and GoM. As stated already in Section 3.1, age is undefined in areas where the released tracer has not yet reached (i.e. where C is equal to zero). When interpreting dye mass fractions, it is important to note that dye end-members are strictly defined by their initial sub-region. For example, GB-dyed water that enters the SS region will remain GB-originating water, no matter whether it first travelled into the GoSL or entered the SS directly.

## 4 Results

### 4.1 Dye Tracer Distributions

The distributions of the vertically averaged dye tracers at 6 months after initialization from the TRANS experiment (Figure 4) show intricate transport patterns on the shelf and indicate that cross-shelf exchange is inhibited in some regions. Notably, neither Slp-D nor Slp-S dye has moved onto the shelf within the first 6 months of the simulation, except for intrusions of Slp-D dye into the deep Laurentian and Northwest Channels. Significant cross-shelf exchange has occurred only at the northernmost region of the Labrador Shelf where a sizeable fraction of LS dye has moved onto the shelf. Another fraction of the LS dye has moved along the shelf break around the Grand Banks and along the edge of the Scotian Shelf, following the 200 m isobath, with a small branch moving into the Gulf of St. Lawrence at the mouth of the Laurentian Channel.

GB dye has moved into the Gulf of St. Lawrence and onto the Scotian Shelf, with larger concentrations on the outer shelf, but has not crossed the shelf break. GB dye has remained the longest in the Laurentian Channel and a portion of GB dye has remained highly localized on the southern tip of the Grand Banks, while the rest of the region is occupied by ENS dye. The distribution of GoSL dye within the Gulf of St. Lawrence also indicates some regions with relatively large retention times, e.g. shallow parts around Prince Edward Island and north of Anticosti Island. A portion of the GoSL dye has moved onto Scotian Shelf. SS dye has moved quickly to the south, but very little has crossed the shelf break. On the Scotian Shelf, the coastal and shelf-break currents have caused the dye to move particularly fast near the coast and just inside of the shelf break, while dye was retained slightly longer mid-shelf. GoM dye has slowly circulated in the Gulf before leaving the domain.

Transects of dye concentrations and currents along the Halifax Line (Figure 1) at the same time and from the same experiment (TRANS) are shown in Figure 5. These transects illustrate that the shelf-break current, visible by the LS dye, acts like a wall along the shelf break inhibiting cross-shelf exchange. Little Slp-D dye travels onto the shelf, and similarly little SS dye travels off the shelf.

### 4.2 Mean Residence Times

Time series of the dye mass leaving their source regions per unit time from the TRANS experiment were used to calculate mean residence times. Two examples, for the Scotian Shelf and Gulf of St. Lawrence, are shown in Figure 6. In general, a

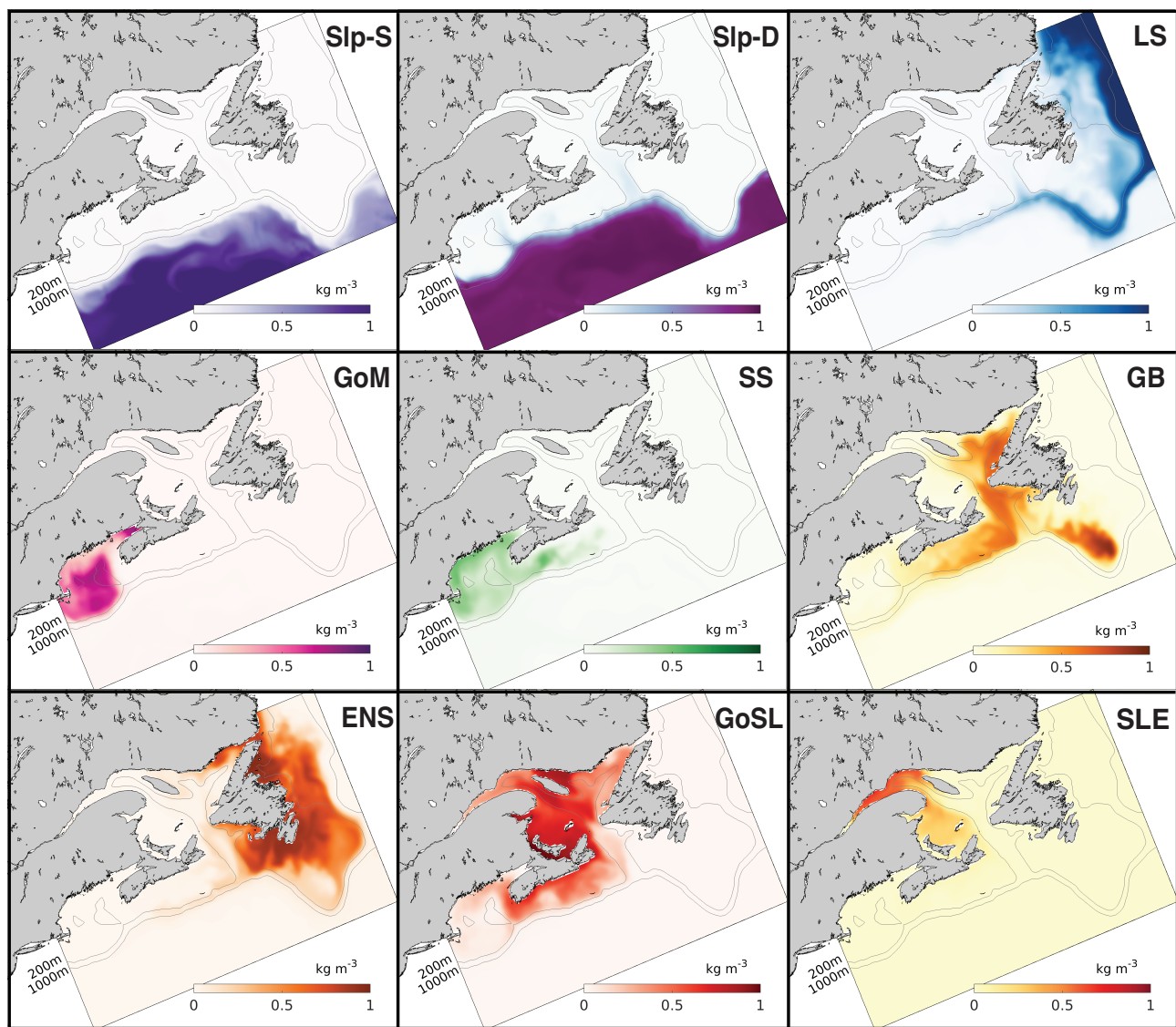

**Figure 4.** Maps of vertical mean dye concentration on June 16, 1999 (6 months into the TRANS simulation).

large pulse of mass leaves each subregion initially, then the amount of mass leaving per unit time decays. SS dye leaves its source region much more rapidly than GoSL dye. The latter displays seasonality that is correlated with the discharge from the St. Lawrence River, when taking into account a lag of 8 months. River discharge peaks in late spring/early summer, with a second, smaller pulse in November. The rate of GoSL mass leaving the Gulf increases in late fall/early winter. The delay of 8 months from peak discharge to peak in mass export from the Gulf is consistent with previous estimates of transport time from the St. Lawrence River to the Scotian Shelf (Sutcliffe et al., 1976; Smith, 1989; Shan et al., 2016), which assumes that the dye

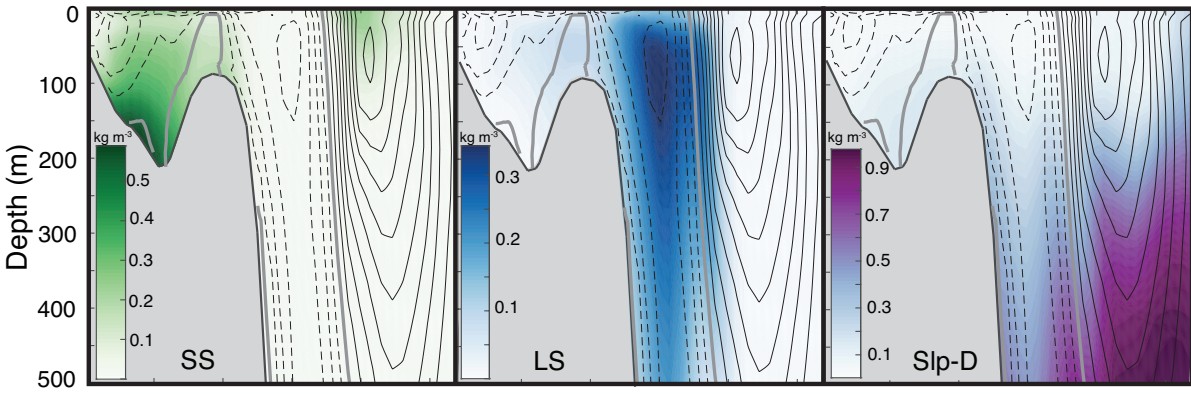

**Figure 5.** Dye concentration (colors) and 5-day average velocities (isolines) along the Halifax Line (HAL) transect on June 16, 1999 (6 months into TRANS simulation). Northward flow (into the page) is indicated by solid lines, zero flow by the thick grey line, and southward flow (out of the page) by dashed lines. Interval between isolines is 0.03 m s$^{-1}$.

is entrained in the river plume. This timing indicates that the seasonal increase in dye mass leaving the Gulf is enhanced by increased river discharge into the Gulf.

MRTs, calculated as the first moment of these histograms (see Equation 12), are 88.1 days for the GB calculated without the Laurentian Channel (126 days with Laurentian Channel), 367 days for the GoSL, 81.4 days for the SS, and 251 days for the GoM. The Scotian Shelf and Grand Banks have the shortest mean residence times, approximately one third of that for the Gulf of Maine and one quarter of that of the Gulf of St. Lawrence. As described in Section 4.1, the SS dye moves quickly to the south, hence mean residence time is short. Similarly, GB dye moves quickly to the south into the Gulf of St. Lawerence and onto the Scotian Shelf, with some dye remaining on a localized patch on the southern tip of Grand Banks. There is a high concentration of GB dye in the Laurentian Channel (Figure 4) indicating a longer retention of GB dye here. The Gulf of Maine and Gulf of St. Lawrence are semi-enclosed basins and more retentive.

### 4.3 Mass Fractions

The AGE simulation was used to calculate the mass fractions from different source regions for selected regions and locations (Figure 8, Table 1). Mass fractions were spun-up for 3 years until they reached a dynamic steady state (Figure 7). Mass fractions reported in Figure 8 and Table 1 were calculated as the average fractions after the 3-year spin up.

Figure 8a shows the mass fraction of each dye contributing to the total dye mass in the Gulf of St. Lawrence, Scotian Shelf and Gulf of Maine regions, not considering the dye initialized in the analyzed region. The main contributions to Scotian Shelf waters are from the GoSL, GB, LS and ENS regions. The main contributions to the Gulf of St. Lawrence are from the ENS, GB and LS regions. The Gulf of Maine has nearly equal contributions from most dyes, except for the SLE dye which makes up only a quarter of the contribution (Table 1; SLE contributing 3% compared to 11-13%), and the SS and GB dyes contributing approximately 50% more than the other dyes (Table 1; SS and GB contributing 18-20% compared to 11-13%).

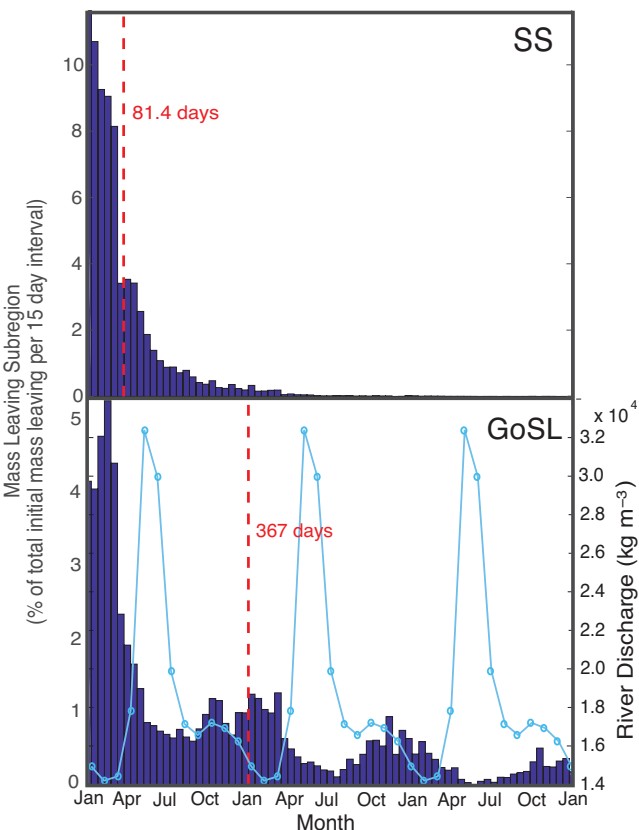

**Figure 6.** Histograms (lifetime distribution) of mass of the SS and GoSL dyes leaving their source regions over time. Mass is reported as a fraction of the initial mass present in each region. Mean residence times are reported in red, calculated as the weighted average of the mass leaving each sub-region per unit time (Equation 12). St. Lawrence River discharge is indicated by the light blue curve in the bottom panel (see right y-axis).

The dye contributions at specific stations on the Scotian Shelf (Figure 8b) illustrate heterogeneity in source-water distributions. For example, the contributions of GoSL dye are high close to shore and minimal near the shelf break. The Nova Scotia Current carries GoSL water along the coast resulting in the high dye contributions at the near-shore stations.

## 4.4  Ages

5    Water ages from the different source regions were calculated, as described in Section 3.1, from the AGE experiment. They are shown as spatial distributions for three selected source regions in Figure 9 and reported as mean ages for the Scotian Shelf and Gulf of St. Lawrence in Table 1. Mean ages are spatially averaged over each sub-region and time averaged over a dynamic steady state unique to each dye. See supplemental figures for timeseries of these mean ages illustrating the spin-up period required to reach a dynamic steady state. Mean ages reflect two important transport timescales: (1) the duration of transport

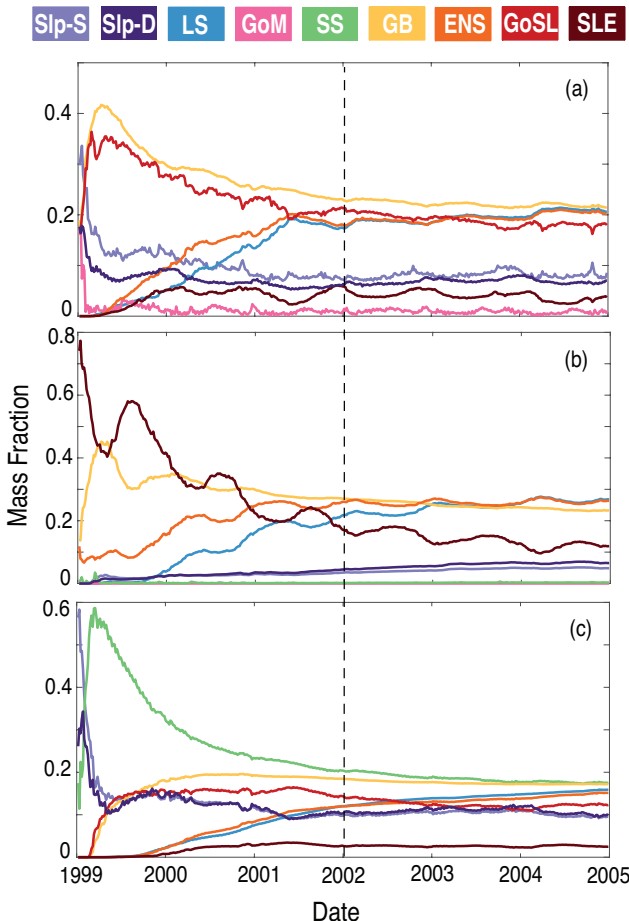

**Figure 7.** Mass fractions of each dye in (a) SS (b) GoSL (c) GoM. Mass fractions are spun-up for 3 years, as noted by the vertical dashed line. The average mass fractions are calculated after this spin-up period.

from source region to the present location, and (2) the mean residence time within the present region. The latter is important when considering mean differences between regions; for example, longer residence times in the Gulf of St. Lawrence contribute to larger mean ages here as compared to the Scotian Shelf. The former contributes to relative differences in ages within a region; for example, on the Scotian Shelf, longer transport times across the shelf contribute to larger LS ages nearshore compared to

5    near the shelfbreak.

The surface age distributions (Figure 9) illustrate the length of transport from a source region to any given location in the model domain and reflect specific transport pathways in the region. GB has low ages on the Scotian Shelf (less than 300 days), indicating the quick transport over the Laurentian Channel to the Scotian Shelf, rather than first entering the Gulf of St. Lawrence. GB dye that first enters the Gulf of St. Lawrence before being transported along the Scotian Shelf via the NSC is

10   found farthest inshore, illustrated by larger ages here. LS water has ages less than 200 days on the East Newfoundland Shelf

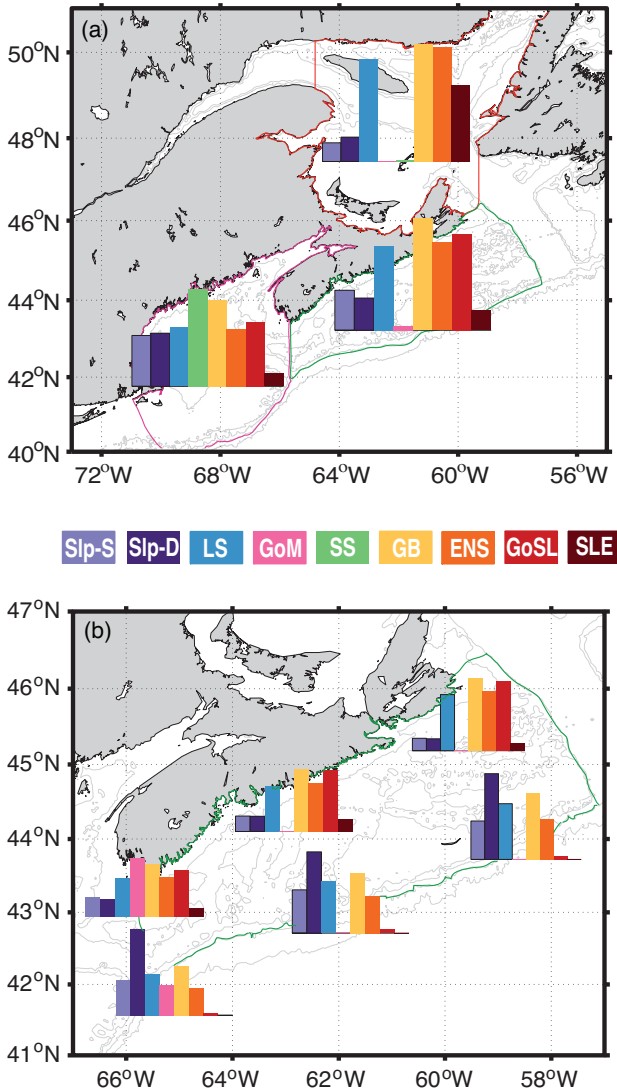

**Figure 8.** (a) Mass fractions of each dye in the three main regions of interest: GoSL (outlined in red), SS (outlined in green), and GoM (outlined in pink). (b) Mass fractions of each dye at six different stations on SS compared to the overall mass fractions of SS (see Table 2 for exact station locations).

and Grand Banks, demonstrating the quick transport onto the shelf here. Although LS ages along the shelf break are relatively small (~200 days), LS water ages are at least three times larger farther inshore on the Scotian Shelf (500-600 days). The large transport times onto the Scotian Shelf are not a result of slow transport south, indicated by small shelfbreak LS ages, but rather a result of LS dye first traveling into the Gulf of St. Lawrence before reaching the inner Scotian Shelf. Slp dye similarly moves easily onto Grand Banks, through the Laurentian Channel, and into Gulf of Maine, but it cannot quickly and directly flow onto the Scotian Shelf. The importance of these transport pathways is illustrated by comparing the dye mass contributions (Figure

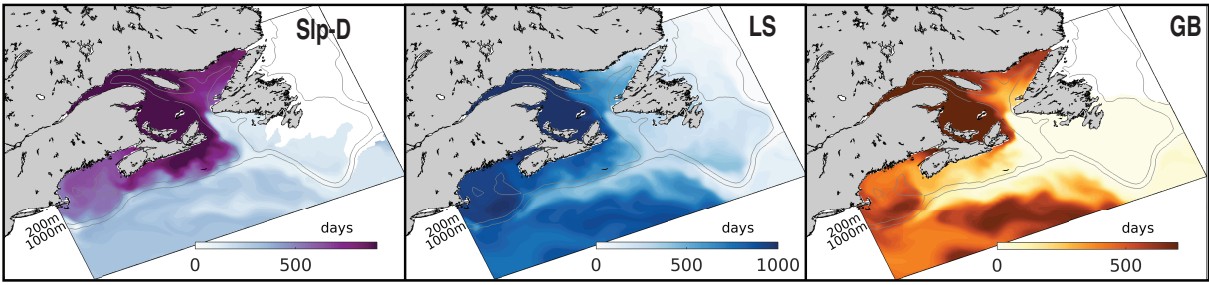

**Figure 9.** Surface age (days) as determined from the AGE experiment. Snapshot from last year of 6 year simulation (June 10, 2004). White areas are regions where dye concentrations $< 0.001$ kg m$^{-3}$.

**Table 1.** Dye mass fractions and mean age (days) of each of the dyes in the shelf regions.

| | | Slp-S | Slp-D | LS | GoM | SS | GB | ENS | GoSL | SLE |
|---|---|---|---|---|---|---|---|---|---|---|
| MASS FRACTION | GoSL | 0.041 | 0.054 | 0.224 | 0.000 | 0.003 | 0.258 | 0.252 | – | 0.168 |
| | SS | 0.084 | 0.067 | 0.177 | 0.009 | – | 0.234 | 0.185 | 0.202 | 0.042 |
| | STN 1 | 0.043 | 0.041 | 0.195 | 0.000 | – | 0.250 | 0.206 | 0.241 | 0.024 |
| | STN 2 | 0.132 | 0.296 | 0.192 | 0.000 | – | 0.229 | 0.139 | 0.012 | 0.000 |
| | STN 3 | 0.058 | 0.058 | 0.175 | 0.000 | – | 0.240 | 0.187 | 0.236 | 0.046 |
| | STN 4 | 0.156 | 0.293 | 0.186 | 0.002 | – | 0.216 | 0.135 | 0.012 | 0.000 |
| | STN 5 | 0.070 | 0.062 | 0.136 | 0.208 | – | 0.187 | 0.142 | 0.164 | 0.031 |
| | STN 6 | 0.129 | 0.320 | 0.150 | 0.110 | – | 0.180 | 0.100 | 0.010 | 0.001 |
| | GoM | 0.106 | 0.110 | 0.122 | – | 0.202 | 0.180 | 0.120 | 0.134 | 0.026 |
| MEAN AGE | GoSL (days) | 1010 | 1030 | 915 | 1750 | 1170 | 639 | 766 | – | 346 |
| | *Standard Deviation* | 67.6 | 67.4 | 62.9 | 57.4 | 82.5 | 47.3 | 56.5 | – | 15.2 |
| | SS (days) | 496 | 648 | 728 | 1140 | – | 206 | 587 | 112 | 324 |
| | *Standard Deviation* | 50.3 | 54.4 | 46.2 | 106 | – | 37.3 | 48.1 | 16.3 | 19.6 |

8) to the mean ages. In each subregion, the waters with the lowest dye contributions tend to have the highest mean ages and vice versa. In the Gulf of St. Lawrence, the LS, GB and ENS dyes have the highest contributions and some of the lowest mean ages: 915, 639, and 766 days, respectively. On the Scotian Shelf, GB and GoSL dyes have the highest contributions and the lowest ages of 206 and 112 days, respectively.

## 5    Discussion

### 5.1    Regional Circulation Features

The simulation of numerical tracers in our model illustrates its accurate representation of circulation features described in previous studies of the region. Circulation in the North Atlantic is dominated by the subpolar gyre, specifically the equatorward transport of the Labrador Current (Loder et al., 1998; Fratantoni and Pickart, 2007). The Labrador Current originates as part of the East Greenland Current, flowing out of the Denmark Strait southward (Fratantoni and Pickart, 2007). The movement of LS dye equatorward in our model simulation, notably following the shelf-break of the Scotian Shelf, represents the dominant movement of Arctic-origin waters southwestward.

The Labrador Current, reinforced by a branch of Cabot Strait outflow, makes up the shelf-break current (Beardsley and Boicourt, 1981; Loder et al., 1998), which agrees with our findings that LS water, combined with GB and GoSL outflow through the Cabot Strait, is a dominant component of the shelf-break current. Previous studies found that Labrador Sea water only penetrates the Gulf of St. Lawrence through the Strait of Belle Isle and through the northeastern part of the Cabot Strait (Galbraith et al., 2013). Our study finds similar results, with limited penetration of the LS dye into the Gulf of St. Lawrence, except through the Laurentian Channel and Strait of Belle Isle. Similarly, it is thought that limited Labrador Sea water penetrates the Scotian Shelf through canyons and gullies along the shelf break (Han, 2003), consistent with the vertical mean dye concentration maps (Figure 4). Studies also found that upper slope flow occurs between the 200 m and 500 m isobaths, consistent with downstream remnants of the shelf-edge Labrador Current (Han et al., 1997; Loder et al., 1998, 2003; Fratantoni and Pickart, 2007). LS water in our model similarly follows the 200 m isobath along the slope. Loder et al. (2003) and Hannah et al. (1996, 2001) report that density fields suggest that there is limited movement of subpolar water southward past the western edge of the Scotian Shelf, which agrees with our results. LS water is young along the shelf break (less than 200 days) but becomes much older once in the Gulf of Maine (over 1000 days). The long transit time of LS dye into the Gulf of Maine suggests that this is not a dominant pathway in the system.

On the Scotian Shelf, past studies have revealed that the dominant circulation pattern is the southwestward flow with seasonal variability in both the Nova Scotia Current (directly along the coast) and the shelf-break current (farther offshore) (e.g. Drinkwater et al., 1979; Han et al., 1997). We note the same dominant circulation structure in our model. GoSL dyes flow through the Cabot Strait onto the Scotian Shelf and, combined with GB and LS dyes as an extension of the Labrador Current, form the noted near-shore and shelf-break southwestward flow. The Slp dyes flow northeastward further offshore.

### 5.2    Mean Residence Times

The mean residence times in the region are: 88.1 days on the Grand Banks, 367 days in the Gulf of St. Lawrence, 81.4 days on the Scotian Shelf and 251 days in the Gulf of Maine. High transport directly along the shelf, with inner-shelf and shelf-break branches (Han et al., 1997), carries SS dye quickly to the southwest into the Gulf of Maine rather than crossing the shelf break. Quantitatively, this quick transport is reflected by relatively short mean residence times on the Scotian Shelf (81.4 days).

Few studies have looked at quantifying retention times on the Scotian Shelf and surrounding regions. Rogers (2015) did, however, look at the retention of particles in shelf basins. Rogers' numerical modeling study of the region used particle tracking to calculate the retention of particles in two basins on the shelf: the Lahave and Emerald Basins. Max retention times were calculated as 166 days in the Lahave Basin and 111 days in Emerald Basin in summer at 200 m release depth. These results, although calculated over different scales and regions of the Scotian Shelf, are similar in magnitude to the mean residence times calculated in our model.

In a larger scale study, Bourgeois et al. (2016) divided the global continental shelves into 43 segments and calculated the residence time in each segment of a global model with an average model resolution of 38 km in our region (ranging from 31 to 45 km). Their Florida Upwelling segment contains the Scotian Shelf, Gulf of Maine and most of the eastern United States shelves. Their Sea of Labrador segment includes the Labrador Sea, East Newfoundland Shelf, Grand Banks and Gulf of St. Lawrence. Bourgeois et al. (2016) reported residence times of $12 \pm 0.6$ and $36 \pm 11$ days for the Florida Upwelling and Sea of Labrador segments, respectively. These residence times are much shorter than the estimates from our model simulations, likely as a result of the coarser resolution ($\sim$38 km vs. $\sim$10 km in ours) missing important circulation features like the shelf-break current. Differences in approaches to calculating residence time may also contribute. Bourgeois et al.'s (2016) residence times are also shorter for other regions when compared to previous studies including the North Sea (Jickells, 1998; Delhez et al., 2004), Baltic Sea (Jickells, 1998), South Atlantic Bight (Jickells, 1998), and the Yellow and East China Seas (Men and Liu, 2014).

Sharples et al. (2017) calculated residence times for $5 \times 5^o$ coastal ocean grid cells around the globe, based on a sequence of theoretical arguments and empirical relationships about ocean-shelf exchange processes. While their analysis does not use subregions as defined in our study, we can make rough comparisons between our model mean residence times and theirs. For the North Atlantic, Sharples et al. (2017) calculated a mean residence time of 14 days for Grand Banks (95% confidence interval (CI): 9-26 days), 105 days for the Gulf of St. Lawrence (95% CI: 62-199 days), 126 days for the Scotian Shelf (95% CI: 74-235 days), and 217 days in the Gulf of Maine (95% CI: 115-509 days). Their estimates for the Scotian Shelf and Gulf of Maine are similar to those calculated here, but are much smaller for the Gulf of St. Lawrence and Grand Banks (approximately a third and a sixth, respectively) than ours. Sharples et al.'s (2017) calculation focuses on river plume export across the shelf break without the use of numerical models. They divide global continental shelves based on the ratio of the plume width to the shelf width, $S_p$, which determines the assumptions needed to calculate residence time. Since the Scotian Shelf has an $S_p < 1$, it is assumed that the salinity plume is confined to the shelf and that exchange with the open ocean is controlled by exchange across the shelf break. It is then assumed that residence time is controlled by Ekman cross-shelf break transport and a lumped transport representing the mean non-wind driven export that can affect residence times on such shelves. Their calculation is therefore very different than the one performed in the present study, resulting in discrepencies in reported residence times.

## 5.3 Transport Times

Since the St. Lawrence River and the Gulf of St. Lawrence have a strong influence on the hydrography of the Scotian Shelf, previous studies have estimated the time for St. Lawrence River discharge to reach Halifax and Cape Sable. Shan et al. (2016)

calculated the lag between the discharge peak from the St. Lawrence River and the low-salinity signal at Halifax and Cape Sable in their model simulations. They found that water exiting the St. Lawrence River took 7 months to travel from Quebec City to Halifax and 8 months to travel to Cape Sable. Sutcliffe et al. (1976) compared the temperature and salinity of the St. Lawrence River to different stations along the Scotian Shelf to determine the time for river discharge to reach these locations. They found, similar to Shan et al. (2016), that St. Lawrence River discharge reaches Halifax in approximately 6 months. Smith (1989) similarly compared the seasonal cycle of current velocity, surface salinity and temperature at different locations and found that St. Lawrence River discharge reached Cape Sable in 8 to 9 months.

In our study, we can compare the age of SLE dye on the Scotian Shelf to these transit estimates, as the age of SLE dye estimates the time for water leaving the estuary to reach specific points on the shelf. We calculated that SLE dye has an average age of 324 days (10.5 months) on the Scotian Shelf, which is longer than in these past studies. Looking at specific locations (Halifax and Cape Sable), the age of SLE dye near Halifax is approximately 9 months (seasonally varies from 6.5 - 12.5 months) and 11.5 months (seasonally varies from 8 - 15 months) near Cape Sable. These estimates are larger than those previously reported but with a similar delay in St. Lawrence River water delivery between Halifax and Cape Sable. For a more direct comparison, we calculated the lag between the St. Lawrence River discharge peak and low-salinity signals at Halifax and Cape Sable in our model simulation, as in Shan et al. (2016). The lag at each of these locations, respectively, was on average 4.8 and 5.5 months, with some interannual variability in lag times. Our calculated lags are smaller than the age estimates but close in magnitude to those from Sutcliffe et al. (1976), Smith (1989), and Shan et al. (2016). The differences between our age calculation, and calculating the lag between peak river discharge and low-salinity signal stems from the fact that the age is a time-integrated measure rather than a propagation of the maximum signal. At certain times of the year, the age of St. Lawrence Estuary water at Halifax, for instance, can reach as low as 6.5 months, which is comparable to previous studies. In other words, it takes longer for river water to reach each of the stations when not during peak river discharge and the age calculation reflects averaging over these times.

## 5.4 Transport Mechanisms

Previous studies have reported that nutrient-rich slope water flows onshore via major cross-shelf channels, such as through the Laurentian Channel, Scotian Gulf and Northeast Channel (Smith et al., 2001). Our study illustrates that the shelf-break current, marked by the LS dye, creates a barrier along the Scotian Shelf that prevents deep slope water from reaching the shelf and similarly prevents surface SS water from being exported to the adjacent open ocean. Small amounts of water marked by LS dye enter the Scotian Shelf through cross-shelf channels, and onshore transport of water marked by Slp dye occurs through the Laurentian Channel and Northeast Channel. However, there is almost no direct onshore transport of Slp-dyed water onto the Scotian Shelf. Slp-dyed water has low ages (Figure 9) and significant mass fractions (Figure 8, Table 1) only near the shelf break; past the mid-shelf ages are much larger and mass fractions lower. We do not find evidence in support for significant upwelling of nutrient-rich slope water onto the Scotian Shelf as suggested by Shadwick et al. (2010) and Burt et al. (2013).

In our simulation, waters marked by GB and GoSL dyes have the highest mass fractions on the Scotian Shelf overall, as indicated by large mass fractions (Figure 8, Table 1) and associated low ages (Figure 9), and are therefore important source

waters for setting the properties on the Scotian Shelf. Although the mean residence time in the Gulf of St. Lawrence is large (367 days), once this water exits the Gulf, it moves quickly along the Scotian Shelf. Similarly, water marked by GB dye moves quickly across the Laurentian Channel onto and along the Scotian Shelf. These pathways have been noted in previous studies. Han (2003) has shown that Gulf of St. Lawrence water moves through the Cabot Strait along the eastern edge of the Scotian

Shelf, while flow from the Newfoundland Shelf directly contributes to Scotian Shelf circulation through crossovers across the Laurentian Channel and along the shelf edge.

## 5.5   Impacts on Regional Zooplankton Distributions

The importance of the NSC, previously estimated to account for 75% of the net annual transport along the shelf (Drinkwater et al., 1979), for setting biological properties on the Scotian Shelf has been investigated in previous studies. For example, Tremblay and Roff

(1983); Sameoto and Herman (1990); Herman et al. (1991); Sameoto and Herman (1992) have evaluated the effects of the NSC on the local copepod community. There are three main species of copepods on the Shelf: *Calanus finmarchicus*, *Calanus glacialis* and *Calanus hyperboreus*. The former dominates the southwestern half of the shelf, whereas the latter two species are dominant on the northeastern half. *C. glacialis* and *C. hyperboreus* are northern species with breeding populations in the cold water of the Gulf of St. Lawrence and Labrador Current, carried to the Scotian Shelf via the NSC and shelf-break cur-

rent (Fleminger and Hulsemann, 1977; Tremblay and Roff, 1983; Conover, 1988). The outflow from the Gulf of St. Lawrence affects the entire northeast portion of the Scotian Shelf (Trites and Walton, 1975; Bugden et al., 1982), acting as a dominant control on the copepod population here (Sameoto and Herman, 1992). *Calanus* species also overwinter in basins deeper than 200-m along the Scotian Shelf (Sameoto and Herman, 1990; Herman et al., 1991) and there is a small contribution from slope waters to shelf *C. finamrchicus* and *C. hyperboreus* populations (Lewis and Sameoto, 1988). By calculating the time for water

to travel from the Gulf of St. Lawrence to the Scotian Shelf, one can estimate the maturity stage of copepods along the shelf, assuming breeding in the Gulf. It is expected that with a 3-month transit time from the Gulf to Halifax copepods will have matured to stage C4 and migrate downwards to the deep basins in the NE shelf (Sameoto and Herman, 1992). It is unlikely that *C. glacialis* and *C. hyperboreus* could exist on the shelf for long without the infusion from the Gulf; *C. finmarchicus* overwinter and are therefore less reliant on the Gulf of St. Lawrence (Sameoto and Herman, 1992). The transport pathways in this region,

mainly the transport of GoSL and GB waters onto the Scotian Shelf to form the NSC and shelfbreak current, are therefore very important controls on setting species populations here.

## 6   Conclusions

In our model simulations, dye tracer distributions and transit timescales are in good agreement with previous modeling and observational studies. Water marked by LS dye flows along the 200 m isobath, forming the shelf-break current which limits the

lateral transport of water on- and off-shelf. Waters marked by LS and Slp flow relatively unimpeded onto the Newfoundland Shelf and Grand Banks, and through the Laurentian and the Northeastern Channels, but do not flow onto the Scotian Shelf. Water marked by LS and Slp dye that is present on the Scotian Shelf has first travelled into the Gulf of St. Lawrence. There

is retention on the tip of the Grand Banks associated with the Labrador Current bifurcating around shallow bathymetry. This retention contributes to a larger mean residence time, compared to the Scotian Shelf. Waters marked by GB and GoSL dyes have the highest contributions on the Scotian Shelf (24% and 20% contributions, respectively) and some of the lowest ages (202 and 113 days respectively). Some of the largest ages on the Scotian Shelf are from waters that must first travel into the Gulf of St. Lawrence before flowing onto the Scotian Shelf. These results reflect the fact that the main pathway onto the Scotian Shelf is first through the Laurentian Channel into the Gulf of St. Lawrence, and then out onto the shelf via the Cabot Strait. In the case of water marked by GB dye, transport occurs directly over the Laurentian Channel.

Our results highlight how useful regional models are in capturing fine-scale details of circulation and quantifying accurate transport times and pathways. Previous estimates of residence time for the NW North Atlantic shelves from a global model are too short, likely because the resolution is too coarse to capture the details of this dynamic system. The mean residence times and transport analyses calculated in this study provide useful information for understanding biological processes, such as the shelf-wide copepod community structure, and biogeochemical processes, such as carbon fluxes.

*Competing interests.* The authors have no competing interests.

*Acknowledgements.* We acknowledge funding by the Marine Environmental Observation Prediction and Response Network (MEOPAR). We are also grateful to two anonymous reviewers for their constructive comments.

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
