# Peer review of "Diagnosing transit times on the northwestern North Atlantic continental shelf"

_Ocean Science, 2018_

## Referee Comment (RC1) · Anonymous Referee #1 · 22 Jun 2018

The authors present a very nice study on water exchange between different subregions of the North Atlantic continental shelf. The chosen method is appropriate, the article is well written, the introduction into the topic is broad, and the conclusions are supported by the results.

There are, however, three major points I would like to see improved:

————————-

(1) As reviewers always do, I request some model validation. You refer to a different article describing the model setup, but that's not sufficient. You need to show, preferably in the online supplement, that your model is able to capture the hydrographic features of the system and its subbasins. What I would expect are vertical profiles of salinity

or density which illustrate whether or not the stratification (as consequence of mixing and estuarine circulation) in the different subbasins is simulated appropriately. Also, a cross-shore transect of salinity from coast to offshore would be very helpful.

(2) One of the main points of your study is emphasizing the weak exchange between coastal and offshore waters. Now a lateral mixing of water masses typically takes place by mesoscale and submesoscale eddies. As your model is not eddy-resolving, I would like to either (a) see a justification whether your model is still able to get the horizontal exchange right, e.g. by comparing mesoscale features like eddies separating from the surface current to satellite observations, or (b) read a paragraph discussing the model limitations stating this as a possible source of error.

(3) There are two ways in which the dye concentrations can be interpreted, and you are mixing them up.

(a) You can use them to indicate the origin of the water, that is, the first region the water parcel resided in. In this case, every water parcel has just one color. That means that you just have to add dye to the "uncolored" water, coming e.g. from rivers or precipitation. In practice it means in the "yellow" area, you increase the concentration of yellow dye until the sum of all dye concentrations is equal to 1 kg/m3. (b) You can use them to indicate all areas the water parcel has travelled through. In this case you increase the yellow dye concentration in the yellow area always to 1 kg/m3, irrespective of the other dyes. The water can be both red and yellow then, indicating it has previously been to two areas.

Interpreting "mass fractions" means you normalize the tracer concentration in such way that the sum of all dyes (maybe except the local one) is equal to one. But this makes sense only in case (a), not in case (b). A simple example can illustrate it:

Think of a single straight river with three areas: upstream=blue, mid-stream=yellow, downstream=red. All of the water arriving downstream originates from upstream and has passed mid-stream, so it has 1 kg/m3 blue and 1 kg/m3 yellow dye in it. Calculating

mass fractions, like you do, means we find 50% blue and 50% yellow dye. But what do these 50% tell us? In fact, they are meaningless. In reality, 100% of the water came from upstream, and also 100% of the water came from mid-stream, this is no contradiction.

If you, of course, only compare the mass fractions against each other, in the sense that one region contributed more than the other (or, in our example, both regions contributed equally), that's correct, but it does not require normalization. I suggest that you leave out normalization in Fig, 7 and 8 and Table 1.
* * *
Apart from these three points, I really like the article and have just a few minor comments:

Page 2, line 34-35: Could you write a few more words about the TTD approach, so a reader not familiar with it can have an idea on how it works?

Page 3, line 10: "it is necessary to describe residence time as a distribution" -> "it is necessary to describe residence time in a finite volume as a distribution"

Figure 1: "the main panel" -> "the upper panel"?

Figure 2: "2 depth levels (200 m and above, and below 200 m)" -> "2 depth levels (above and below 200 m)"

Page 5, line 3: "an age tracer" -> in Deleersnijder et al., this is called "age concentration tracer" to indicate that it does not store the age, but rather the product between age and concentration. I would suggest using this wording throughout the manuscript to avoid confusion.

Page 5, line 4-5: "the time since the associated dye tracer has left its initialization region" -> add "for the last time" (it may have left it before and then returned to it)

Page 6, line 16: C(tau,x) should be C(t,x)

Page 8, line 12: "was found to perform better than the MPDATA advection scheme" ->
I do not think this comparison is required, but if you want to compare, please state how
you found out which one performs better.

Page 8, line 13-19: Please explain this a bit more clearly - you use both the Urrego-
Blanco and Sheng 2012 model and the Geshelin et al. climatology as boundary condi-
tions?

Page 9, line 31: "consistent with previous estimates" -> please add references already
here

Figure 4: The values seem rather low. For example, the initial value at SLP-S should
be 200 kg/m2, after six months it has reduced to below 6 kg/m2? Also they do not
seem to match the values shown in Figure 5, if you integrate the concentrations given
there vertically. Also, dashed lines indicating the source region boundaries would be
nice.

Page 10, line 1: "Sotian Shelf"

Page 10, line 1: Please note that the river water discharged during the simulation is
uncolored, so your interpretation requires entrainment of dye into the river plume.

Page 10, line 1-2: "This timing indicates that the seasonal increase in dye mass leaving
the Gulf is driven by increased river discharge into the Gulf." -> I am aware that you did
not give the following simplifying explanation, but a non-oceanographer could easily
misinterpret your sentence: It is certainly not the volume of river discharge pushing
the dye out - it is too small for that. I would rather suppose that the river discharge
enhances the estuarine circulation leading to a better exchange with the open sea.
However, other factors like wind might have a seasonality as well, so the attribution to
the river discharge is not straightforward.

Section 4.4: Please give the day when ages are evaluated in the main text, not just
in the figure caption. How do the mean ages you found relate to the length of your

simulation - did they already reach a dynamic steady state or will they increase if you simulate longer?

Table 1: Stations should show up in a map, e.g. in a slightly larger version of Fig. 2.

Page 17, line 15: What different assumptions did they make compared to your study? Or is it just the resolution?

––––––––––––––––––––––––––––––––

---

## Referee Comment (RC2) · Anonymous Referee #2 · 17 Jul 2018

General comments

The paper "Diagnosing transit times on the northwestern North Atlantic continental shelf" by Krysten Rutford and Katja Fennel is a study aiming to detect retention times, mean ages and transport pathways of water-masses of different origin in the northwestern North Atlantic using two passive tracers: dye tracer and age tracer.

In principle the paper is well written and can be already suggested for publication in "Ocean Science" with minor/moderate revision of current state.

Major comments/suggestions and questions

The major concern is the selection of the time period for the analysis. Namely, is there some sort of inter-annual variability in the circulation system that can somehow

change the results? For example the authors have not found evidence of strong up-welling events in the region, which have previously been indicated by other authors (e.g. Shadwick et al. 2010 and Burt et al. 2013). According to Shadwick et al. (2010) and references therein, coastal Scotian shelf is a well known for coastal up-welling events and these have been successfully produced also by modelling studies (e.g. Donohue, 2000). In this study, the authors did not find any evidence of upwelling induced transport. Why is that?

Second concern is associated with the first one: namely, if the selected period did not have any upwelling events in the region, perhaps the study should be extended for longer period to have full view of the circulation in the region. Nevertheless, if there were upwelling events during the selected period, but the model was unable to re-produce them, perhaps the global atmospheric forcing (ERA-Interim) should be replaced by some regional product, which might have better spatial resolution and also better representation of the local weather climate – the wind patterns for example.

Third concern is also somehow associated with the first one: namely, the tempera-ture and salinity are nudged towards climatology in the open boundaries. This should remove inter-annual variability of temperature and salinity at the boundaries, but how large is the latter?

Minor comments/suggestions and questions

1. Use chronological order of references in the text.

2. Page 2, section "Introduction", lines 13-16: Authors state that for the region this is the first study of residence times, transport pathways and timescales in the NW North Atlantic. Nevertheless, for discussion, they have found several studies, which to compare their results to. Therefore, I would add general statement about other studies.

3. Please state explicitly if you are using ERA-Interim forcing instead of too gen-eral statement in page 8 lines 9-11: . . . surface forcing from the European Centre

for Medium-Range Weather Forcasts (ECMWF) global atmospheric reanalysis Dee et al. (2011) . . .

4. The location of the stations used for histograms in Figure 8b could also be shown in Figure 1.

5. The number format in Table 1 could be consistent – there is no need for scientific notation and I recommend replacing scientific notation with decimal notation.

6. In Figures 4 and 9 the initial location of tracers could be shown by shading the geographic area or drawing solid contours.

7. Page 17, line 15: be more precise with the origin of the differences with Sharples et al. (2017). The statement is too general.

8. Can dye and age tracer leave the model region i.e. are open boundaries used also for those tracers?

---

## Author Comment (AC1) · 13 Aug 2018

**Response to Comments by Reviewer 1**
(Reviews are included in black font; Responses are in blue font)

The authors present a very nice study on water exchange between different subregions of the North Atlantic continental shelf. The chosen method is appropriate, the article is well written, the introduction into the topic is broad, and the conclusions are supported by the results.

**Response:** Thank you for the kind words and for the thorough and helpful review. Below we describe in detail how we intend to address your comments in the revised manuscript.

There are, however, three major points I would like to see improved:

(1) As reviewers always do, I request some model validation. You refer to a different article describing the model setup, but that's not sufficient. You need to show, preferably in the online supplement, that your model is able to capture the hydrographic features of the system and its subbasins. What I would expect are vertical profiles of salinity or density which illustrate whether or not the stratification (as consequence of mixing and estuarine circulation) in the different subbasins is simulated appropriately. Also, a cross-shore transect of salinity from coast to offshore would be very helpful.

**Response:** Agreed. We would like to add transects showing model-data comparisons of temperature and salinity across the Scotian Shelf. More specifically, we will add comparisons of the model to the two Atlantic Zone Monitoring Program (AZMP) transects available for the Scotian Shelf (the Halifax Line and the Louisbourg Line) and glider transects, both of which encompass the cross-shelf T&S gradient and the T&S signals in the deeper shelf basins. We will additionally include a time series of Scotian Shelf area-averaged temperature and salinity at the sea surface in comparison to a climatology. These plots will be added to the supplementary information with a summary of a few other key points from the Brennan et al. (2016) paper that provides a detailed validation of our physical model, especially how our model set-up reproduces the volume transport in the region as compared to Loder et al.'s (1998) estimated mean annual transport.

(2) One of the main points of your study is emphasizing the weak exchange between coastal and offshore waters. Now a lateral mixing of water masses typically takes place by mesoscale and submesoscale eddies. As your model is not eddy-resolving, I would like to either (a) see a justification whether your model is still able to get the horizontal exchange right, e.g. by comparing mesoscale features like eddies separating from the surface current to satellite observations, or (b) read a paragraph discussing the model limitations stating this as a possible source of error.

**Response:** With a horizontal resolution of 10 km, we would argue that our model is eddy-resolving, although it certainly is not resolving sub-mesoscale variability. Shown below are selected snapshots of the model-simulated surface temperature, which illustrate clearly that a

range of mesoscale circulation features is produced, such as tendrils of cool water off the Newfoundland and Nova Scotia coasts, and meanders in the Gulf Stream.

[Figure]

[Figure]

[Figure]

**temperature**

Oct-17-2004

200m
1000m

(3) There are two ways in which the dye concentrations can be interpreted, and you are mixing them up.

(a) You can use them to indicate the origin of the water, that is, the first region the water parcel resided in. In this case, every water parcel has just one color. That means that you just have to add dye to the "uncolored" water, coming e.g. from rivers or precipitation. In practice it means in the "yellow" area, you increase the concentration of yellow dye until the sum of all dye concentrations is equal to 1 kg/m3.

**Response:** In our experiments at model initialization, every water parcel has just one color equal to 1 kg/m3. We use the term "water parcel" here to mean the volume of a single grid box, which is the smallest volume we can practically consider. As the simulation progresses, water from different source regions mixes, and thus each grid cell can have a mixture of different dyes. We are not sure why the Reviewer states that "every water parcel has just one color" because clearly many grid cells have water from multiple source regions and thus different dyes. Perhaps the Reviewer is thinking of Lagrangian particles? Those would be different.

(b) You can use them to indicate all areas the water parcel has travelled through. In this case you increase the yellow dye concentration in the yellow area always to 1 kg/m3, irrespective of the other dyes. The water can be both red and yellow then, indicating it has previously been to two areas.

**Response:** Agree, this is exactly what is happening in our AGE simulations.

Interpreting "mass fractions" means you normalize the tracer concentration in such way that the sum of all dyes (maybe except the local one) is equal to one. But this makes sense only in case (a), not in case (b). A simple example can illustrate it:

Think of a single straight river with three areas: upstream=blue, mid-stream=yellow, downstream=red. All of the water arriving downstream originates from upstream and has passed mid-stream, so it has 1 kg/m3 blue and 1 kg/m3 yellow dye in it. Calculating mass fractions, like you do, means we find 50% blue and 50% yellow dye. But what do these 50% tell us? In fact, they are meaningless. In reality, 100% of the water came from upstream, and also 100% of the water came from mid-stream, this is no contradiction.

**Response:** We can't entirely follow the Reviewer's logic here. Below we apply to Reviewer's thought experiment to our two different types of simulations.

In the TRANS simulation (used to calculated residence times): In a **purely advective** river, where the three different water types do not mix, and the dyes are initialized as indicated by the Reviewer, the water arriving at the mouth would be either only red (initially), only yellow (once the mid-stream water has propagated to the river mouth), or only blue (once all the mid-stream water has propagated through and the upstream water is arriving). All the concentrations would be 1 kg/m$^3$. **But** if the system has **advection and mixing**, then water parcels will have mixtures of the different dyes and the dye concentrations will be equal to or smaller then 1 kg/m$^3$.

In the AGE simulation (used to get mass fractions and ages): In the **purely advective** case, initially only red dye would arrive at the river mouth, later both red dye (at 1 kg/m$^3$) and yellow dye (at 1 kg/m$^3$) would arrive simultaneously, and finally red dye (at 1 kg/m$^3$), yellow dye (at 1 kg/m$^3$) and blue dye (1 kg/m$^3$) would all arrive simultaneously. **But** in the case with **advection and mixing**, initially only red dye would arrive, later red dye (at 1 kg/m$^3$) and yellow dye (at < 1 kg/m$^3$) would arrive simultaneously and finally red dye (at 1 kg/m$^3$), yellow dye (at < 1 kg/m$^3$) and blue dye (< 1 kg/m$^3$) would all arrive simultaneously.

The only case where the water can have "1 kg/m$^3$ blue and 1 kg/m$^3$ yellow dye" at the river mouth is the purely advective AGE case. If there is mixing, this cannot occur in either the AGE or the TRANS cases. And our model does include mixing. In our opinion it is meaningful to report the dye mass fractions.

We agree with the Reviewer on his/her statement: "In reality, 100% of the water came from upstream, and also 100% of the water came from mid-stream, this is no contradiction."

If you, of course, only compare the mass fractions against each other, in the sense that one region contributed more than the other (or, in our example, both regions contributed equally), that's correct, but it does not require normalization. I suggest that you leave out normalization in Fig, 7 and 8 and Table 1.

**Response:** This is indeed why we report mass fractions – we want to illustrate which region contributed more and which contributed less. We prefer to use the normalization, because this yields more manageable ranges in the numbers we report.

Apart from these three points, I really like the article and have just a few minor comments:

Page 2, line 34-35: Could you write a few more words about the TTD approach, so a reader not familiar with it can have an idea on how it works?

**Response:** We would like to updates this sentence to read: "TTD is best used for steady flow applications and computes the full spectrum or distribution of transit times in a water parcel using Green's functions (Haine and Hall 2002), while CART is better suited to time-varying flow and is especially useful for highly resolved coastal applications."

Page 3, line 10: "it is necessary to describe residence time as a distribution" -> "it is necessary to describe residence time in a finite volume as a distribution"

**Response:** Agree, will be changed as suggested.

Figure 1: "the main panel" -> "the upper panel"?

**Response:** Agree, will be changed as suggested.

Figure 2: "2 depth levels (200 m and above, and below 200 m)" -> "2 depth levels (above and below 200 m)"

**Response:** We defined the first depth level to include 200 m (i.e. >=200 m) whereas the deeper depth level is below 200m (<200 m), so would prefer to keep it as is.

Page 5, line 3: "an age tracer" -> in Deleersnijder et al., this is called "age concentration tracer" to indicate that it does not store the age, but rather the product between age and concentration. I would suggest using this wording throughout the manuscript to avoid confusion.

**Response:** Will be changed as suggested.

Page 5, line 4-5: "the time since the associated dye tracer has left its initialization region" -> add "for the last time" (it may have left it before and then returned to it)

**Response:** Agree and would like to change to: "the time since the associated dye tracer has **last** left its initialization region"

Page 6, line 16: C(tau,x) should be C(t,x)

**Response:** Thank you for noticing this typo. Will be changed as suggested.

Page 8, line 12: "was found to perform better than the MPDATA advection scheme" -> I do not think this comparison is required, but if you want to compare, please state how you found out which one performs better.

**Response:** Comparison will be removed.

Page 8, line 13-19: Please explain this a bit more clearly - you use both the Urrego- Blanco and Sheng 2012 model and the Geshelin et al. climatology as boundary conditions?

**Response:** Urrego-Blanco and Sheng's 2012 model is used as boundary conditions, the Geshelin climatology is used in the climatology file for weak nudging inside the domain, but not at the boundary. This will be clarified in the manuscript text.

Page 9, line 31: "consistent with previous estimates" -> please add references already here

**Response:** Agreed, we will add reference to Shan et al. (2016), Sutcliffe et al. (1976), and Smith et al. (1989) here.

Figure 4: The values seem rather low. For example, the initial value at SLP-S should be 200 kg/m2, after six months it has reduced to below 6 kg/m2? Also they do not seem to match the values shown in Figure 5, if you integrate the concentrations given there vertically. Also, dashed lines indicating the source region boundaries would be nice.

**Response:** Thank you for catching this mistake. There was an error in the code for plotting that will be updated. We will also implement your suggestion to include the source region boundaries in both Figure 4 and Figure 9.

Page 10, line 1: "Sotian Shelf"

**Response:** Thank you for catching this typo. Will be corrected.

Page 10, line 1: Please note that the river water discharged during the simulation is uncolored, so your interpretation requires entrainment of dye into the river plume.

**Response:** We will add a sentence indicating that we assume dye is entrained in the river plume.

Page 10, line 1-2: "This timing indicates that the seasonal increase in dye mass leaving the Gulf is driven by increased river discharge into the Gulf." -> I am aware that you did not give the following simplifying explanation, but a non-oceanographer could easily misinterpret your sentence: It is certainly not the volume of river discharge pushing the dye out - it is too small for that. I would rather suppose that the river discharge enhances the estuarine circulation leading to a better exchange with the open sea. However, other factors like wind might have a seasonality as well, so the attribution to the river discharge is not straightforward.

**Response:** We suggest changing the wording from "dye mass leaving the Gulf is driven by increased river discharge…" to "…dye mass leaving the Gulf is enhanced by increased river discharge…"

Section 4.4: Please give the day when ages are evaluated in the main text, not just in the figure caption. How do the mean ages you found relate to the length of your simulation - did they already reach a dynamic steady state or will they increase if you simulate longer?

**Response:**

In response to part 1 of this comment: The in-text ages are from dynamic steady state and are therefore averaged over the steady-state period (the period of dynamic steady state varies between each individual dye). An explanation will be added to the text.

In response to part 2 of this comment: Ages have already reached a dynamic steady state, illustrated by our supplementary information. We previously ran our model for 9 years to ensure the mean ages reported would not increase with a longer simulation. A couple of the dye mean ages might increase slightly with increased simulation length but the supplemental figures show that most of the dyes have already reached a steady cycle with minimal increase expected.

Table 1: Stations should show up in a map, e.g. in a slightly larger version of Fig. 2.

**Response:** We will update Figure 2 to include the station locations.

Page 17, line 15: What different assumptions did they make compared to your study? Or is it just the resolution?

**Response:** The calculation itself as well as the objectives in Sharples et al. (2017) study are very different from ours. They focus on calculating export of river plume water to the open ocean without the help of numerical models based only on a few simple assumptions. One step in their calculation is the estimation of residence times. To accomplish this goal, they have divided the global shelves into two different cases using the $S_p$ ratio (i.e. the ratio of the plume width to the shelf width). If $S_p > 1$, the low salinity plume is assumed to reach beyond the shelf edge into the open ocean and therefore the residence time is assumed to be governed by cross-shelf transport mechanisms within the buoyant plume. If $S_p < 1$, the low salinity plume is assumed to be confined to the shelf within the buoyancy current. River plume water is assumed to gradually mix with shelf water and therefore exchange with the open ocean should be controlled by transport across the shelf break. The Scotian Shelf falls into this latter category with $S_p < 1$. The residence time is therefore calculated as a function of the mean shelf depth, the shelf width and the transports that are assumed to dominate in this scenario ($S_p < 1$), which are cross-shelf break Ekman transport and a lumped transport that estimates the nonwind-driven exchange.

We will add in a short description explaining this.

---

## Author Comment (AC2) · 13 Aug 2018

**Response to Comments by Reviewer 2**
(Reviews are included in black font; Responses are in blue font)

General comments:
The paper "Diagnosing transit times on the northwestern North Atlantic continental shelf" by Krysten Rutherford and Katja Fennel is a study aiming to detect retention times, mean ages and transport pathways of water-masses of different origin in the northwestern North Atlantic using two passive tracers: dye tracer and age tracer.

In principle the paper is well written and can be already suggested for publication in "Ocean Science" with minor/moderate revision of current state.

**Response:** We are grateful for the positive assessment and constructive review. Below we a detailed response to all comments and describe how we intend to address them in the revised manuscript.

Major comments/suggestions and questions:

The major concern is the selection of the time period for the analysis. Namely, is there some sort of inter-annual variability in the circulation system that can somehow change the results? For example the authors have not found evidence of strong upwelling events in the region, which have previously been indicated by other authors (e.g. Shadwick et al. 2010 and Burt et al. 2013). According to Shadwick et al. (2010) and references therein, coastal Scotian shelf is a well known for coastal upwelling events and these have been successfully produced also by modelling studies (e.g. Donohue, 2000). In this study, the authors did not find any evidence of upwelling induced transport. Why is that?

**Response:**
With regard to the length of the simulation: Our model was run for 9 years in the AGE simulations and thus should capture a sufficiently long period for the results not to be unduly influenced by interannual variability.

With regard to upwelling: We would like to clarify the important distinction between coastal upwelling (typically within 10 km of the coast) and upwelling of deep water along the shelf break (at about 200 km from the coast). Our dye tracer experiments can only be used to evaluate upwelling of deep slope water (from below 200 m in the slope region) at the shelf break, but not coastal upwelling in which Scotian Shelf water from below the seasonal thermocline upwells in the vicinity of the coast. In summer, winds can be southwesterly along the coast of Nova Scotia, which is the upwelling-favourable direction and this frequently leads to coastal upwelling. Petrie et al. (1987) used satellite images of the region to show the development of a band of cool water along the southern shore of Nova Scotia over the month of July 1984 caused by upwelling-favourable winds (see Figure below). The modeling study by Donohue (2000) reproduced the event studied by Petrie et al. This coastal upwelling event occurred, as stated by Petrie et al. (1987), "over a coastal strip about 10 km wide and 500 km long."

[Figure]

**Figure 1: Satellite infrared imagery of sea surface temperatures from (a) July 7, (b) July 14, (c) July 21, (d) July 25, (e) July 31 and (f) August 6, 1984. Image is from Petrie et al. (1987) illustrating narrow band of cool water on the southern shore of Nova Scotia during a period of upwelling-favourable winds.**

A more recent example from Shan (2016) showing both satellite images and simulated model snapshots of SST in July 2012 is given below and illustrates again the band of cool upwelled waters

on the southern shore of Nova Scotia in the vicinity of the coast. Shan (2016) noted two distinct upwelling events during 2012, one that peaked July 22 and the other September 1, 2012. Shadwick et al. (2010) and Burt et al. (2013) did not show any direct evidence of upwelling; instead they invoked it as an explanation of their carbon observations.

[Figure]

**Figure 2: MODIS satellite remote sensing data of SST and Chlorophyll concentrations over the central Scotian Shelf and adjacent waters from July 22 and September 1, 2012 (from Shan 2016). Note that the shelf break is outside the frames. 100 m and 200 m isobaths are shown in black and gray contour lines, respectively.**

[Figure]

**Figure 3: Snapshots of simulated SST over the central Scotian Shelf in July 2012 with instantaneous wind stress vectors plotting as black arrows (DalCoast-CSS model from Shan 2016).**

Our model produces coastal upwelling events similar to those observed. The snapshots of our model simulation below show the narrow band of cool upwelled water along the southern coast of Nova Scotia.

We emphasize again that the shelf break is ~ 200 km from shore and propose adding some text to the manuscript to clarify these distinct types of upwelling.

[Figure]

References:

Donohue, S. M. A numerical model of an upwelling event off the coast of Nova Scotia, MSc thesis, Royal Military College of Canada, Kingston, ON, 2000.

Petrie, B., B. Topliss, and D. Wright, Coastal upwelling and eddy development off Nova Scotia, *Journal of Geophysical Research*, 92, 12979-12991, 1987.

Shan, S. Eulerian and Lagrangian studies of circulation on the Scotian Shelf and adjacent deep waters of the North Atlantic with biological implications, PhD thesis, Dalhousie University, Halifax, NS, 2016.

Second concern is associated with the first one: namely, if the selected period did not have any upwelling events in the region, perhaps the study should be extended for longer period to have full view of the circulation in the region. Nevertheless, if there were upwelling events during the selected period, but the model was unable to re-produce them, perhaps the global atmospheric forcing (ERA-Interim) should be replaced by some regional product, which might have better spatial resolution and also better representation of the local weather climate – the wind patterns for example.

**Response:** As stated above, our model does indeed capture the **coastal** upwelling that occurs on the Scotian Shelf as a result of upwelling-favourable winds, therefore we do not believe we need to consider replacing our atmospheric forcings.

Third concern is also somehow associated with the first one: namely, the temperature and salinity are nudged towards climatology in the open boundaries. This should remove inter-annual variability of temperature and salinity at the boundaries, but how large is the latter?

**Response:** It is true that there is no interannual variability in the boundary conditions; however, coastal upwelling is driven primarily by wind forcing which does vary interannually. The model does display interannual variability in its simulated coastal upwelling.

Minor comments/suggestions and questions:

1. Use chronological order of references in the text.

**Response:** This will be updated.

2. Page 2, section "Introduction", lines 13-16: Authors state that for the region this is the first study of residence times, transport pathways and timescales in the NW North Atlantic. Nevertheless, for discussion, they have found several studies, which to compare their results to. Therefore, I would add general statement about other studies.

**Response:** We will update these lines to the following: "Although previous studies have quantified shelf basin particle retention (Rogers 2015), shelf residence times as part of global studies (Bourgeois et al. 2016, Sharples et al. 2017), and transport times from the St. Lawrence River to the Scotian Shelf (Sutcliffe et al. 1976, Smith 1989, Shan et al. 2016), this is the first comprehensive analysis of residence times, transport pathways and timescales in the NW North Atlantic."

3. Please state explicitly if you are using ERA-Interim forcing instead of too general statement in page 8 lines 9-11: . . . surface forcing from the European Centre for Medium-Range Weather Forcasts (ECMWF) global atmospheric reanalysis Dee et al. (2011) . . .

**Response:** We will update to specify that it is ECMWF ERA-Interim forcing

4. The location of the stations used for histograms in Figure 8b could also be shown in Figure 1.

**Response:** We will add the station locations to either Figure 1 or Figure 2.

5. The number format in Table 1 could be consistent – there is no need for scientific notation and I recommend replacing scientific notation with decimal notation.

**Response:** We will update and remove the scientific notation.

6. In Figures 4 and 9 the initial location of tracers could be shown by shading the geographic area or drawing solid contours.

**Response:** We will add the location of the initial dye tracer regions to Figure 4 and 9.

7. Page 17, line 15: be more precise with the origin of the differences with Sharples et al. (2017). The statement is too general.

**Response:** We will add a sentence detailing some of the specifics from Sharples et al. (2017). Specifically, we will emphasize that their focus is on calculating river plume export across the shelf break without numerical models and that they divide global continental shelves based on Sp ratio (whether Sp < 1 or Sp > 1 will determine the assumptions used to calculate residence time). Since the Scotian Shelf has an Sp <1 it is assumed that the salinity plume is confined to the shelf and that exchange with open ocean is controlled by exchange across the shelf break. Residence time is therefore assumed to be controlled by Ekman cross-shelf break transport as well as a lumped transport that factors in mean non-wind-driven export that can affect residence times on such shelves.

8. Can dye and age tracer leave the model region i.e. are open boundaries used also for those tracers?

**Response:** Yes, the boundaries are open for dye and age tracers and they can therefore leave the model domain.